# Urea Delays High-Temperature Crosslinking of Polyacrylamide for In Situ Preparation of an Organic/Inorganic Composite Gel

**DOI:** 10.3390/gels11040256

**Published:** 2025-03-31

**Authors:** Li Liang, Junlong Li, Dongxiang Li, Jie Xu, Bin Zheng, Jikuan Zhao

**Affiliations:** 1Key Laboratory of Optic-Electro Sensing and Analytical Chemistry of Life Science, Ministry of Education, Qingdao University of Science and Technology, Qingdao 266042, China; tliang_li@163.com (L.L.); lijunlong0307@163.com (J.L.); xujie@qust.edu.cn (J.X.); zhengbin@qust.edu.cn (B.Z.); 2Lab of Colloids and Functional Nanostructures, College of Chemistry and Molecular Engineering, Qingdao University of Science and Technology, Qingdao 266042, China

**Keywords:** polyacrylamide, crosslinking reaction, composite gel, urea, delayed gelation, in situ preparation

## Abstract

To address the rapid crosslinking reaction and short stability duration of polyacrylamide gel under high salinity and temperature conditions, this paper proposes utilizing urea to delay the nucleophilic substitution crosslinking reaction among polyacrylamide, hydroquinone, and formaldehyde. Additionally, urea regulates the precipitation of calcium and magnesium ions, enabling the in situ preparation of an organic/inorganic composite gel consisting of crosslinked polyacrylamide and carbonate particles. With calcium and magnesium ion concentrations at 6817 mg/L and total salinity at 15 × 10^4^ mg/L, the gelation time can be controlled to range from 6.6 to 14.1 days at 95 °C and from 2.9 to 6.5 days at 120 °C. The resulting composite gel can remain stable for up to 155 days at 95 °C and 135 days at 120 °C. The delayed gelation facilitates longer-distance diffusion of the gelling agent into the formation, while the enhancements in gel strength and stability provide a solid foundation for improving the effectiveness of profile control and water shut-off in oilfields. The urea-controlling method is novel and effective in extending the high-temperature cross-linking reaction time of polyacrylamide. By converting calcium and magnesium ions into inorganic particles, it enables the in situ preparation of organic/inorganic composite gels, enhancing their strength and stability.

## 1. Introduction

The gelation time, stability duration, and mechanical strength of polymer gels are pivotal parameters for their successful application in oilfield profile control and water shut-off treatments [1,2,3,4,5,6,7]. These parameters directly influence the duration and effectiveness of on-site production operations. However, high temperatures can accelerate gelation, while high salinity conditions can drastically shorten the stability duration of polymer gels, presenting ongoing challenges for researchers [8,9,10,11]. Simply adjusting the concentrations of polymers and crosslinking agents has proven insufficient for effectively delaying gelation at elevated temperatures [12,13,14,15], with limited literature addressing the modulation of crosslinking reactions based on fundamental reaction mechanisms. To overcome these limitations and enhance gel performance, organic/inorganic composite gels (CG) have attracted significant attention [2,4,6,16,17,18,19,20,21,22,23,24,25,26,27,28,29,30,31,32,33,34,35]. The integration of organic and inorganic components within composite gels leverages various interactions including electrostatic forces, hydrogen bonding, and coordination bonds to markedly improve the strength and stability of the resulting materials [16,17,18,19]. In addition, inorganic particles can also serve as multifunctional crosslinkers to form physical crosslinking points with polymer molecules. Organic/inorganic composite gels exhibit improved hydrophilicity, thermal stability, salt resistance, and higher elastic modulus, and have become an important research direction for enhanced oil recovery [20,21,22,23,24,25,26,27,28,29,30,31,32,33,34,35]. A wide array of inorganic particles, including silica [16,19,20,21,22], titanium dioxide [23], alumina [24], zirconia [25], fly ash [26], Fe_3_O_4_ [27], montmorillonite [28,29,30], bentonite [31], laponite [17,32], magnesium aluminum layered double hydroxide [33], zirconium hydroxide [18], graphene oxide [23,34], micron graphite oxide powder [35], and carbon nanotubes [23], have been explored for this purpose. However, incorporating these inorganic particles often requires regulating particle size [19,20], layer thickness [28,30,33], and surface properties [29,31] to ensure compatibility. Some systems struggle with high-salinity formation waters, necessitating the use of low-concentration brines or freshwater for gel preparation [17,19,34], which limits practical applications. The in situ synthesis of organic/inorganic composite gels from high-salinity polymer solutions remains underexplored. It is well-known that divalent metal ions such as Ca^2+^ and Mg^2+^ in formation brine can catalyze polymer hydrolysis, leading to over-crosslinking and syneresis at high temperature [18,36,37]. There is a need for scientific approaches to utilize mineral resources within formations for the effective in situ preparation of composite gels.

Significant research has explored the crosslinking reaction mechanisms involving polyacrylamide (PAM), hydrolyzed polyacrylamide (HPAM), binary and ternary copolymers containing acrylamide [1,4,5,6,7,12,13,14,15,38,39,40,41,42,43]. Organic crosslinkers that either contain or can produce formaldehyde are frequently utilized, including formaldehyde itself, hexamethylenetetramine (HMTA), phenol/catechol/resorcinol/hydroquinone (HQ)-formaldehyde or HMTA, and melamine-formaldehyde, etc., [12,13,14,15,38,39,40,42,43,44]. The reaction mechanisms of these crosslinking systems typically fall into two categories [42]. The first involves hydroxymethylation reactions between formaldehyde and PAM or phenolic substances, leading to the introduction of hydroxymethyl groups on the amide nitrogen and ortho/para positions of phenolic hydroxyls, followed by condensation dehydration to form a three-dimensional network [1,4,43]. The second category encompasses hydroxymethylation reactions between formaldehyde and phenolic or melamine molecules, introducing hydroxymethyl groups that subsequently react with amide groups in the polymer through condensation, ultimately forming a robust three-dimensional gel structure [12,13,14,40].

HMTA, which gradually decomposes into formaldehyde and ammonia above 80 °C [14], has emerged as a more environmentally friendly alternative to formaldehyde. Studies [10,13,44] have examined the gelation performance of PAM/HMTA/HQ and HPAM/HMTA/methyl p-hydroxybenzoate crosslinking systems across varying pH levels. The reaction system can form a stable gel when the pH is maintained between 7.5 and 9.5. However, gelation does not occur when the pH exceeds 9.5 [10]. By lowering the pH of the reaction system from 7.9 to 4.5 using an acid solution, the gelation time is significantly reduced from 10.6 h to 3.2 h, and the gel strength increases from 0.053 MPa to 0.062 MPa [13]. It has been proposed that an acidic environment promotes the decomposition of HMTA into formaldehyde, which further accelerates the crosslinking reaction, leading to a faster reaction rate and increased crosslink density. In contrast, an alkaline environment inhibits the decomposition of HMTA and the subsequent crosslinking reactions [5,13]. Notably, in gel systems involving HMTA, the decomposition of HMTA generates ammonia gas, which reacts with water to dissociate and release hydroxide ions, thereby increasing the system’s alkalinity. Consequently, when analyzing the impact of pH on gelation performance, the primary focus has been on how acidity and alkalinity influence the decomposition of HMTA, which is considered the main factor affecting the gelation performance of the reaction system. However, the direct effect of pH on the crosslinking reaction itself has often been overlooked.

Bryant et al. [15] investigated the gelation performance of acrylamide and the acrylamide-based AM-AMPS binary copolymer within a phenol-formaldehyde crosslinking system, with a particular focus on the influence of pH. In this system, formaldehyde acts directly as a reactant, and its quantity or concentration remains independent of the solution’s pH. The experiments demonstrated that gelation occurs within a pH range of 1.0 to 8.5, with the crosslinking reaction proceeding rapidly at pH 1, while the rate of crosslinking is the slowest under neutral conditions. Feng et al. [39] utilized ^13^C-NMR spectroscopy to explore the reaction process between PAM and formaldehyde. The study revealed that at a high pH (9), formaldehyde reacts with PAM to produce N-hydroxymethylated polyacrylamide without leading to crosslinking. Conversely, at low pH (1), formaldehyde can react with PAM to form methylene-bis-acrylamide crosslinked structures. These findings indicate that pH plays a significant regulatory role in the crosslinking reaction between formaldehyde or phenolic formaldehyde crosslinking agents and acrylamide polymers.

Marandi et al. [38] explored the crosslinking reaction of formaldehyde with PAM catalyzed by hydrochloric acid at pH 5. Under acidic conditions, the protonation of the hydroxymethyl group facilitates the nucleophilic attack of amide nitrogen atoms on the carbon atoms of the hydroxymethyl groups, resulting in the formation of a gel structure with methylene diacrylamide serving as the crosslinking point. The crosslinking reaction proceeds via a nucleophilic reaction mechanism. In another study, melamine/formaldehyde (MF) resins rich in hydroxymethyl groups were used to crosslink HPAM [40], with optimal pH levels for stable, high-strength gels ranging from 5 to 9. Outside this range, gel viscosity decreases and strength weakens due to polymer chain coiling and hydrolysis. It is suggested that the reaction between the amide groups and the hydroxymethyl groups in the MF resin follows a nucleophilic substitution reaction mechanism [40]. Acidic conditions promote the protonation of hydroxymethyl groups, thereby enhancing the crosslinking between amide groups and hydroxymethyl groups. Conversely, an increase in the pH of the reaction system inhibits the protonation of hydroxymethyl groups, which impedes the nucleophilic attack of the nitrogen atom in the amide group on the carbon of the hydroxymethyl group, ultimately preventing the establishment of a dense crosslinked network structure within the system.

Although a pH range of 5–9 is favorable for the formation of high-strength gels, acidic conditions can cause corrosion of the internal walls of the wellbore. To mitigate this issue, ammonium chloride was utilized as a catalyst to regulate the crosslinking reaction between HPAM and MF at pH levels of 8–9 [40]. At a temperature of 80 °C, this approach successfully reduced the gelation time from 7 days to a mere 8 h while simultaneously enhancing gel strength. It is suggested that the ammonium ions generated from the ionization of ammonium chloride donate protons to the hydroxymethyl groups, thereby accelerating the crosslinking reaction within the described system [40]. This study not only validates the nucleophilic substitution mechanism of the reaction between PAM and organic crosslinkers but also provides valuable insights for regulating gelation time based on this mechanism. References [45,46] have documented the use of ammonium salts to accelerate polymer gelation, particularly to facilitate the crosslinking reaction between polyacrylamide and phenolic formaldehyde compounds under low-temperature conditions. These findings will contribute to the rapid sealing of formations and the protection of oil and gas reservoirs during production activities.

Based on the nucleophilic substitution crosslinking reaction mechanism between PAM and phenolic formaldehyde compounds [38,40], this research employs urea additives to regulate both the gelation time and strength of the composite gel. Urea decomposes at elevated temperatures, producing ammonia and carbon dioxide, which further generate hydroxide and carbonate anions in the solution. These anions inhibit the protonation of the methylol group, thereby extending the gelation time of the gelant. In high-salinity solutions, divalent calcium and magnesium ions interact with carboxylate ions produced from the hydrolysis of amide groups through electrostatic or coordination bonds. Additionally, these calcium and magnesium ions can also react with carbonate or hydroxide ions generated from the decomposition of urea to form inorganic particles, such as carbonates or basic carbonates. These particles subsequently bind closely with polymer molecules, resulting in the formation of high-strength composite gels. This study utilizes divalent metal ions in formation brine as raw materials to in situ synthesize organic/inorganic composite gels. This approach not only mitigates the destabilizing impact of inorganic salts on polymer gels but also enhances the strength and stability of the composite gel through the incorporation of generated inorganic particles, thereby achieving integrated utilization of subsurface mineral resources.

## 2. Results and Discussion

In this study, urea was employed to regulate the crosslinking reaction between polyacrylamide and hydroquinone/formaldehyde crosslinkers, as well as the precipitation reactions of calcium and magnesium ions. The amount of urea was carefully controlled to maintain a specific molar ratio between urea and the divalent metal ions in the solution, denoted as *N* = *n*_urea_/(*n*_Ca_^2+^ + *n*_Mg_^2+^), where *n*_urea_, *n*_Ca_^2+^ and *n*_Mg_^2+^ are the amounts of substance of urea, calcium ions, and magnesium ions, respectively. The gelling solution or gelant transforms into an organic/inorganic composite gel (CG) with defined viscoelastic properties after high-temperature aging in the oven. For performance comparisons, freshwater gel (FG) and brine gel (BG) samples without urea were also prepared. In this paper, the samples of freshwater gel, brine gel, and composite gel are designated as FG-t, BG-T_x_-t, and CG-T_x_-U_N_-t, respectively, where t signifies the aging temperature of the sample in degrees Celsius (°C), T_x_ indicates that the total dissolved solids or salinity in the gelant equals to x × 10^4^ mg/L, and U_N_ represents the molar ratio *N* between urea and the sum of calcium and magnesium ions in the reaction system.

### 2.1. FT-IR and XRD

The FT-IR spectra of the PAM polymer and FG sample are presented in Figure 1a. The absorption peaks observed at 3443, 1641, and 1104 cm^−1^ correspond to the stretching vibrations of υ_N-H_, υ_C=O_, and υ_C-N_, respectively [9,18,40,47,48]. Furthermore, the peaks at 2920, 2851, 1453, and 1390 cm^−1^ represent the asymmetric stretching, symmetric stretching, asymmetric bending, and symmetric bending absorption peaks of the C–H bonds in the methylene group [9,18,47,48]. These findings confirm the presence of the primary functional groups in polyacrylamide. In the FT-IR spectrum of PAM FG sample, two new absorption peaks at 1570 and 1231 cm^−1^ appear, which are attributed to the stretching vibrations of carbon–carbon conjugated double bonds and the C–O bond stretching vibrations in phenolic compounds, respectively [47,49]. These FT-IR data suggest that a gel structure is formed through a crosslinking reaction between hydroquinone, formaldehyde, and PAM via covalent bonding.

X-ray diffraction (XRD) is an effective technique for analyzing the structure of inorganic materials. In this study, XRD tests were conducted on inorganic particles formed by the reaction of Ca^2+^ and Mg^2+^ ions with hydrolysis products of urea, with the results illustrated in Figure 1b. The characteristic peaks observed at 2θ values of 26.1°, 27.4°, 33.4°, 36.1°, 38.7°, and 46.0° correspond to the (111), (021), (012), (200), (022), and (221) crystal plane diffractions of CaCO_3_ (JCPDS#75-2230) [50]. A prominent peak at 2θ of 30.1° is attributed to the (104) crystal plane diffraction of (Mg_0.129_Ca_0.871_)CO_3_ (JCPDS#86-2336) [51]. The peaks at 2θ values of 12.0°, 15.2°, and 23.7° can be indexed as the (020), (100), and (040) crystal plane diffractions of Mg_7_(CO_3_)_5_(OH)_4_·24H_2_O (JCPDS#47-188) [52]. The XRD results indicate that Ca^2^⁺ and Mg^2^⁺ ions in brine can react with OH^−^ and CO_3_^2−^ generated from urea hydrolysis to form various inorganic particles, including calcium carbonate, magnesium calcium carbonate, and basic magnesium carbonate. The diffraction peak intensities of calcium carbonate and magnesium calcium carbonate are relatively strong, while the intensity of the basic magnesium carbonate peak is lower, suggesting that the crystal structures of the former two inorganic compounds are more well defined, whereas basic magnesium carbonate exhibits significantly greater amorphicity [52]. The equations involving urea decomposition and its products further interacting with calcium and magnesium ions are detailed in Formulas (1) to (7) [53,54].

Urea decomposition reaction equations:CO(NH_2_)_2_ + H_2_O = CO_2_↑ + 2NH_3_↑(1)(2)$${\mathrm{NH}}_{3}+H_{2}O⇌{\mathrm{NH}}_{3}\cdot H_{2}O⇌{\mathrm{NH}}_{4^{+}}+{\mathrm{OH}}^{-}$$



(3)
$${\mathrm{CO}}_{2}+H_{2}O=H_{2}{\mathrm{CO}}_{3}⇌H^{+}+{\mathrm{HCO}}_{3^{-}}$$





(4)
$${\mathrm{HCO}}_{3^{-}}+{\mathrm{OH}}^{-}⇌{\mathrm{CO}}_{3^{2-}}+H_{2}O$$



Precipitation reaction equations for Ca^2^⁺ and Mg^2^⁺ ions:CO_3_^2−^ + Ca^2+^→CaCO_3_(5)CO_3_^2−^ + 0.129Mg^2+^ + 0.871Ca^2+^→ (Mg_0.129_ Ca_0.871_)CO_3_(6)5CO_3_^2−^ + 7Mg^2+^ + 4OH^−^ + 24H_2_O→ Mg_7_(CO_3_)_5_(OH)_4_·24 H_2_O(7)

The amide groups in PAM undergo hydrolysis at elevated temperatures, leading to the formation of carboxyl groups, which subsequently dissociate into carboxylate ions that interact with cations in the solution through electrostatic or coordination forces [1,8,36,55]. The electrostatic interactions between Ca^2^⁺ and Mg^2^⁺ ions and carboxylate ions in the solution are significantly stronger than those with monovalent cations. Additionally, divalent cations can bind to carboxylate ions from different molecular chains, resulting in crosslinking that reduces the hydrophilicity of the polymer, and may even lead to excessive crosslinking and syneresis [1,36,55]. This phenomenon is a major contributor to the reduced stability of the polymer under high mineralization conditions. In PAM solutions with a high concentration of Ca^2+^ and Mg^2+^ ions, the availability of carboxylate ions is relatively limited, meaning that only a fraction of the divalent ions can interact with carboxylate ions through electrostatic or coordination interactions. At elevated temperatures, urea decomposes to produce NH_3_ and CO_2_. As the concentration of NH_3_ increases within the reaction system, it further reacts with water to yield NH_4_^+^ and OH^−^, resulting in an increase in the system’s alkalinity. Concurrently, CO_2_ can generate carbonate ions (CO_3_^2−^) under alkaline conditions. The cations Ca^2+^ and Mg^2+^ that associate with carboxylate ions can precipitate with OH^−^ and CO_3_^2−^ on the surface of the polymer, leading to the formation of inorganic particles such as calcium carbonate, magnesium carbonate, and basic magnesium carbonate. This process resembles biomineralization [56]. These inorganic particles interact with the carboxylate ions of the polymer through electrostatic or coordination forces, resulting in the formation of an organic/inorganic composite gel. Figure 1 illustrates the deposition of calcium and magnesium inorganic particles on the surface of polyacrylamide.

### 2.2. SEM

The experimental results reveal that both the BG sample (BG-T_15_-95, Figure 2a) and CG sample (CG-T_15_-U_1.00_-95, Figure 2d) exhibit a three-dimensional network morphology. During the freeze-drying process, a substantial amount of inorganic salts precipitates continuously on the surface of the crosslinked polymer network in both gels. Figure 2b,e display the SEM images of the BG and CG samples after being washed with water, respectively. While the gel samples maintain their three-dimensional network structure, the surface of the polymer gel becomes smoother after washing, indicating the removal of water-soluble inorganic salts. Upon further magnification of the electron microscope images, it was observed that no significant particles were attached to the surface of the polymer in the BG sample (Figure 2c). In contrast, numerous solid particles remained adhered to the surface of the polymer in the CG sample (Figure 2f), with an average particle size of approximately 80–100 nm. This morphological characteristic is similar to that of the PAM/PEI/CAF composite gels [26]. It is inferred that the solid particles attached to the polymer surface in the composite gel correspond to inorganic particles of calcium carbonate, magnesium carbonate, or basic magnesium carbonate, as indicated by the XRD test results. These insoluble substances, formed from calcium and magnesium ions in conjunction with the decomposition products of urea, cannot be removed through water washing. Moreover, electrostatic or coordination interactions between divalent cations and the carboxylate anions of the polymer facilitate the stable adhesion of the particles to the polymer surface. Consequently, the crosslinked polymer forms an organic/inorganic composite gel with the inorganic particles.

PAM is a linear polymer that undergoes a crosslinking reaction involving its amide groups when exposed to hydroquinone formaldehyde crosslinking agents, resulting in the development of a three-dimensional gel network. When a substantial amount of sodium chloride, calcium chloride, and magnesium chloride is dissolved in water, a three-dimensional brine gel is formed. According to the salinity composition of the brine gel, a certain amount of urea is introduced into the reaction system. Under high-temperature conditions, urea decomposes and facilitates the precipitation reaction of divalent cations, leading to the formation of inorganic particles such as calcium carbonate, magnesium carbonate, and basic magnesium carbonate. These inorganic components interact with the organically crosslinked polymer, resulting in the creation of a three-dimensional composite gel. The precipitation reaction consumes a significant amount of divalent metal ions, thereby reducing their concentrations in the liquid phase and effectively mitigating their adverse effects on the stability of the polymer gel. Concurrently, the inorganic particles are uniformly distributed within the crosslinked polymer matrix, significantly enhancing the elastic modulus, strength, and stability of the composite gel [16,17,26,28].

### 2.3. Gelation Performance

The gelation process and stability of gel samples were recorded based on the gel strength codes proposed by Sydansk et al. [8,57,58,59]. The classification of gel strength levels is detailed in Table 1. The gel strength developments of FG, BG, and CG samples throughout the gelation process at temperatures of 95 °C and 120 °C, are presented in Table 2 and Table 3. Concentrations of PAM (*C*_PAM_), HQ (*C*_HQ_), and HCHO (*C*_HCHO_) for the gel samples are also given. The FG can achieve a maximum strength of I at 95 °C without syneresis. At 120 °C, FG can also reach the strength of I level, but begins to dehydrate after being heated for 126 days. BG samples similarly attain their maximum strength I at 95 °C. However, brine gels exhibit a highest strength of H and these samples are more susceptible to dehydration at 120 °C. CG samples demonstrate the capability to reach a peak strength of I at both 95 °C and 120 °C. Moreover, the CG samples show superior stability under higher urea proportion conditions (*N* = 1.0 − 2.0).

The time required for the gel sample to attain a strength code of D was recorded as the gelation time (*t*_D_) [58,59]. The stability time (*t*_S_) is defined as the duration needed for the gel to dehydrate by 5 mL from a 20 mL gel sample. This work conducts a comparative analysis of the effects of various factors, including urea concentration (*C*_urea_), degree of mineralization, and temperature, on both *t*_D_ and *t*_S_.

*t*_D_ and *t*_S_ data of FG and BG samples are shown in Table 4. The experimental results indicate that the FG samples are characterized by a longer stability time, exceeding 176 days and 126 days at 95 °C and 120 °C, respectively. For the BG samples, as the salinity increases from 5 × 10^4^ mg/L to 20 × 10^4^ mg/L, both the gelation time and stability time are significantly shortened. For example, at 95 °C, the *t*_D_ of BG decreases from 12.4 days to 7.9 days, while the *t*_S_ drops from 42.0 days to 24.5 days. Notably, BG samples exhibit significantly shorter stability times compared to FG samples. When comparing high-temperature samples with low-temperature samples, both gelation time and stability time are shorter for the former. These findings suggest that high temperature and high salinity are the primary factors contributing to the reduced stability of polymer gels [1,8,36,37,55]. Under high-temperature conditions, the thermal motion of polymer and crosslinking agent molecules intensifies, providing higher energy that enables more molecules to participate in the crosslinking reaction, thereby shortening the gelation time [10,16,26]. Under high mineralization conditions, divalent calcium and magnesium ions facilitate polymer hydrolysis, converting amide groups into carboxyl groups, which can further dissociate into carboxylate ions and hydrogen ions. Divalent cations link with carboxylate ions of different polymer chains through ionic interactions [55], while the dissociated hydrogen ions promote nucleophilic substitution crosslinking reactions between amide groups and hydroquinone formaldehyde crosslinking agents. These reactions result in over-crosslinking and syneresis within the system, significantly diminishing the stability of the gel samples [36,37,55,60].

At 95 °C, we investigated the effects of *C*_PAM_, *C*_HQ_, and *C*_HCHO_ on *t*_D_ and *t*_S_ of the composite gel, which had a salinity of 15 × 10^4^ mg/L and a fixed *C*_urea_ of 0.68 wt% (*N* = 0.5). The experimental data are summarized in Table 5, where *C*_PAM_, *C*_HQ_ and *C*_HCHO_ are maintained ranging from 0.5 to 2.0 wt%, 0.06 to 0.31 wt%, and 0.07 to 0.33 wt%, respectively. Under conditions of low polymer concentration (*C*_PAM_ = 0.5 wt%) and low crosslinker concentrations (*C*_HQ_ = 0.06 wt%, *C*_HCHO_ = 0.07 wt%), the stability times of the composite gels are found to be 37.8 days and 69.2 days, respectively. However, the structures of these gel samples remain incomplete and susceptible to dehydration, as their maximum gel strengths correspond to the codes of H and F, rather than the code of I. In contrast, when the PAM concentration is excessively high (1.5–2.0 wt%), the gelation time is significantly reduced to 0.7–1.2 days, resulting in over-crosslinking, syneresis, and a loss of stability. Consequently, the optimized concentrations for *C*_PAM_, *C*_HQ_ and *C*_HCHO_ at 95 °C are determined to be 1 wt%, 0.20 wt%, and 0.21 wt%, respectively. At 120 °C, the preferred *C*_PAM_ remains at 1 wt%, while both *C*_HQ_ and *C*_HCHO_ are adjusted to 0.16 wt% to delay the crosslinking reaction.

To further investigate the effects of salinity and *C*_urea_ on the gelation performance of the composite gel, we examined urea-containing crosslinking systems with *N* values of 0.5 and 1.0 across salinities ranging from 5 to 20 × 10^4^ mg/L. The gelation time and stability performance of the composite gels are presented in Table 6. Due to the proportional amount of urea and divalent cations in the reaction solution, *C*_urea_ increases along with salinity, leading to a gradual extension of the gelation time for the composite gel. When *N* is set at 0.5, an increase in salinity results in an extended gelation time from 2.1 days to 5.2 days, with stability times ranging from 72.1 to 85.2 days, significantly surpassing those of the brine gel. When the *N* value is increased to 1.0, as salinity rises from 5 to 20 × 10^4^ mg/L, the gelation time extends from 2.2 days to 7.0 days, while the stability time of the composite gel further increases to over 155.0 days. These findings underscore the critical role of urea in delaying the crosslinking reaction between polyacrylamide and the hydroquinone formaldehyde crosslinker at elevated temperatures, thereby enhancing the stability of the composite gel.

This paper systematically investigates the effects of *C*_urea_ on the gelation and stability times of a polymer solution with a salinity of 15 × 10^4^ mg/L and a PAM concentration of 1 wt% at temperatures of 95 °C and 120 °C, as illustrated in Figure 3. An increase in urea concentration results in gelation times ranging from 2.5 to 14.1 days at 95 °C and from 1.4 to 6.5 days at 120 °C (Figure 3a), demonstrating that urea effectively regulates the gelation time *t*_D_ of the composite gel. Although the *t*_D_ at 120 °C is shorter than that at 95 °C, higher urea concentrations can extend the *t*_D_ by 3.6 to 4.6 times, leading to significantly longer gelation times for CG samples compared to FG and BG specimens.

In both laboratory experiments and field production, a common approach to prolong gelation time is to adjust the concentrations of crosslinking agents and polymers. For instance, references [12,13,14] indicate that adjustable gelation time ranges are 21–41 h at 100.8 °C, 6.3–15.5 h at 110 °C, and 7–14 h at 150 °C. Although these regulatory measures demonstrate some effectiveness, the overall findings highlight that short gelation times and limited controllable ranges at elevated temperatures remain significant technical challenges. A comprehensive understanding of the crosslinking reaction mechanism, coupled with precise regulation of the concentrations of reaction components, is crucial for effectively extending the high-temperature crosslinking reaction time of the polymer.

In this paper, urea is employed as a base source. At high temperatures, it decomposes into ammonia and carbon dioxide. This process leads to the formation of weak electrolytes, such as hydrated ammonia and carbonic acid, which partially dissociate into NH_4_^+^, OH^−^, H^+^, HCO_3_^−^ and CO_3_^2−^ ions. The anions interact with NH_4_^+^ and H^+^ ions through electrostatic forces, effectively suppressing the protonation of hydroxymethyl groups and thereby prolonging the crosslinking reaction time at elevated temperatures [40]. The ionization equilibrium of the weak electrolyte shifts forward only when Ca^2+^ and Mg^2+^ ions in the solution consume a sufficient amount of OH^−^ and CO_3_^2−^ ions, allowing NH_4_^+^ and H^+^ ions to facilitate the crosslinking reaction. Based on the nucleophilic reaction mechanism, the gelation time of the PAM solution can be extended, thereby creating favorable conditions for deep profile control and water plugging in oilfields.

Figure 3b illustrates the correlation between the stability time *t*_S_ of the composite gel and the urea concentration *C*_urea_. At 95 °C, it is clear that as *C*_urea_ increases from 0.14 wt% (*N* = 0.1) to 0.41 wt% (*N* = 0.3), the stability time of the composite gel extends from 36.0 days to 55.7 days. Further increasing *C*_urea_ to 0.68 wt% (*N* =0.5) and 1.02 wt% (*N* = 0.75) results in a significant increase in stability time, reaching 78 days and 122 days, respectively. Notably, when *C*_urea_ is elevated to the range of 1.35–2.71 wt% (*N* = 1.0–2.0), the stability time can extend up to 155 days. At 120 °C, the stability time of the composite gel is slightly reduced compared to that at 95 °C. Specifically, as *C*_urea_ increases from 0.14 wt% (*N* = 0.1) to 1.02 wt% (*N* = 0.75), the stability time gradually rises from 9.75 days to 31.6 days. This trend suggests that high-temperature conditions may lead to over-crosslinking and syneresis, which can reduce stability time. However, further increasing *C*_urea_ significantly prolongs the stability time of the composite gel at 120 °C. For example, when *C*_urea_ increases from 1.35 wt% (*N* = 1.0) to 2.71 wt% (*N* = 2.0), the stability time of the composite gel samples can reach 135 days.

Urea facilitates the precipitation of Ca^2+^ and Mg^2+^ ions, leading to the formation of carbonate particles. This process not only effectively reduces the concentration of divalent cations that undermine gel stability but also enhances the stability of the composite gel by incorporating inorganic particles. Under the influence of urea, the stability time of the composite gel significantly increases, allowing it to maintain strength at I or H levels for an extended period under high-temperature and high-salinity conditions (Table 2 and Table 3). This stability is crucial for improving production efficiency in oilfields.

### 2.4. Dynamic Rheological Property

It is well established that during the aging process of the gelant, urea gradually decomposes, and the alkaline environment effectively inhibits hydroxymethyl protonation, thereby extending the gelation time at elevated temperatures. Concurrently, the OH^−^ and CO_3_^2−^ anions present in the reaction system can precipitate with Ca^2+^ and Mg^2+^ ions in highly mineralized solutions, leading to the formation of inorganic particles that subsequently create organic/inorganic composite gels with the crosslinked polymers. To analyze the impact of the composite structure on gel strength, this study evaluated the dynamic rheological properties of the gel samples. The elastic modulus and viscous modulus of FG, BG, and CG samples were measured, and the effects of urea concentration and salinity on gel performance were assessed, as illustrated in Figure 4.

Figure 4a illustrates the elastic modulus and the viscous modulus curves of FG, BG, and CG samples. The composite gels were prepared under varying urea concentration conditions, with a salinity of 15 × 10^4^ mg/L. All samples were aged at 95 °C for 44.7 days. Firstly, the overall experimental results indicate that the elastic modulus G′ of all gel samples is significantly greater than the viscous modulus G”, demonstrating that the gel networks formed through crosslinking reactions possess excellent deformation characteristics and quasi-solid properties [21,31,61], which aid in plugging large pores in the formation and controlling the water absorption profile. Secondly, under experimental conditions with oscillation frequencies ranging from 0.01 to 10 Hz, the elastic modulus G′ of the freshwater gel FG-95 is the lowest (45.3–55.6 Pa), followed by the brine gel BG-T_15_-95 (51.3–73.7 Pa), while the composite gel exhibits the highest elastic modulus (Figure 4a). It can be analyzed that, although the polymer molecular chains in the FG system are the most extended, the strength of the pure organic gel is the lowest. In contrast, the BG containing inorganic salts demonstrates a higher elastic modulus than the FG under the same conditions, attributed to calcium and magnesium ions promoting the hydrolysis of PAM and crosslinking with the formed carboxylate groups [1,8,36], thereby enhancing the gel strength. When urea is introduced into the reaction solution, the inorganic particles in the prepared composite gel interact with the polymers through electrostatic or coordination forces, and this composite structure imparts a higher elastic modulus to the product [16,17]. It is evident that both the crosslinked structure and the composite structure directly influence the enhancement of gel strength. Thirdly, the elastic modulus of the composite gel increases with rising urea concentration. In Figure 4a, there are six CG samples including CG-T_15_-U_0.30_-95, CG-T_15_-U_0.50_-95, CG-T_15_-U_0.75_-95, CG-T_15_-U_1.00_-95, CG-T_15_-U_1.25_-95, and CG-T_15_-U_1.50_-95. The salinity and polymer concentration were fixed at 15 × 10^4^ mg/L and 1 wt%, respectively. As the *N* value increases from 0.30 to 1.50, the elastic modulus ranges for the six CG samples are 64.5–75.6 Pa, 60.0–82.4 Pa, 75.2–90.9 Pa, 94.9–131.2 Pa, 103.5–144.0 Pa, and 103.8–150.3 Pa, respectively. These results indicate that the elastic modulus of CG-T_15_-U_0.30_-95 is slightly higher than that of BG-T_15_-95 (51.3–73.7 Pa). As the *N* value further increases to 0.50 and 0.75, the strengths of these two composite gel samples rise, showing an increase of 10–15 Pa in elastic modulus compared to that of CG-T_15_-U_0.30_-95. When the *N* value reaches 1.00 in sample CG-T_15_-U_1.00_-95, there is a significant increase in gel strength, with an increase of about 20–40 Pa. When the *N* value is further increased to 1.25–1.50, the elastic modulus G′ of the composite gel continues to show an upward trend, but the increase is relatively small. Under low urea concentration conditions (*N* = 0.3 − 1.0), as the urea concentration increases, the reaction between calcium and magnesium ions in the composite gel and the hydrolysis products of urea can produce a larger number of inorganic particles, which helps to increase the strength of the composite gel [17,21,26]. Under high urea concentration conditions (*N* = 1.25 − 1.50), further increasing the urea content creates an alkaline environment due to urea decomposition, which is unfavorable for the PAM crosslinking reaction. As shown in Figure 3a, high urea concentration conditions can significantly prolong the gelation time of the composite gel. Therefore, at *N* = 1.25 − 1.50, the increase in the elastic modulus of the composite gel diminishes.

Figure 4b presents the elastic modulus G′-oscillation frequency curves of three urea-containing composite gels (CG-T_15_-U_1.00_-120, CG-T_15_-U_1.25_-120, and CG-T_15_-U_1.50_-120) aged 7.3 days at 120 °C, with *N* values of 1.00, 1.25, and 1.50, respectively. Under the experimental conditions, the elastic modulus ranges of these three composite gels are 40.2–54.8 Pa, 41.1–56.6 Pa, and 41.7–57.8 Pa, respectively. As the *N* value increases from 1.00 to 1.50, the elastic modulus of the composite gels increases slightly. This observation is similar to the phenomenon noted in Figure 4a. The influence of urea on the growth of inorganic particles and its effect on the PAM crosslinking reaction jointly determine the elastic modulus of the composite gel. In Figure 4b, the elastic modulus of the CG sample CG-T_15_-U_1.00_-120 (120 °C, aged for 7.3 days, *C*_PAM_ = 1.0 wt%, *C*_HQ_ = *C*_HCHO_ = 0.16 wt%) is lower than that of the CG sample CG-T_15_-U_1.00_-95 depicted in Figure 4a (95 °C, aged for 44.7 days, *C*_PAM_ = 1.0 wt%, *C*_HQ_ = 0.20 wt%, *C*_HCHO_ = 0.21 wt%). When the composition of the gel agent is the same, high temperature promotes the polymer crosslinking reaction and the formation of inorganic particles, resulting in a composite gel with a faster gelation rate and higher gel strength. To delay the crosslinking reaction of the composite gel under high temperature conditions of 120 °C, we selected a lower concentration of crosslinking agent to prolong the gelation time and control the gel strength. Additionally, since the aging time of the composite gel at 120 °C is relatively short in this study, the elastic modulus of the composite gel samples in Figure 4b is relatively small. In fact, delaying the gelation rate of the gel system under high-temperature conditions and controlling the increase in gel strength is more advantageous for the gelant to penetrate deeply into the formation. Furthermore, as indicated in Table 3, the CG samples under high-temperature conditions demonstrate long-term stability, with gel strength reaching level I.

Figure 4c depicts the elastic modulus of composite gel samples, i.e., CG-T_5_-U_1.00_-95, CG-T_10_-U_1.00_-95, CG-T_15_-U_1.00_-95, and CG-T_20_-U_1.00_-95. As salinity and urea concentration increase, the elastic modulus of the composite gel increases correspondingly. For salinities of 5, 10, 15, and 20 × 10^4^ mg/L, the elastic modulus ranges for the CG samples are 68.0–95.1 Pa, 77.0–114.2 Pa, 94.9–131.2 Pa, and 119.1–156.4 Pa, respectively. These results confirm the role of inorganic particles formed in brine in enhancing the elastic modulus of the composite gel. Increased urea concentrations and salinities result in a higher quantity of inorganic particles in the system, thus strengthening the composite gel [17,21,26]. This study employs a precipitation reaction between products of urea decomposition and calcium and magnesium ions to treat inorganic salts, converting destabilizing metal ions into beneficial inorganic particles. Simultaneously, these generated inorganic particles are used to construct the composite gel in situ with organic crosslinked polymers, thereby improving its elastic modulus. This process exemplifies an effective integration of inorganic salt treatment and application, converting adverse factors that destabilize the gel into favorable conditions that improve its strength and stability. This research has introduced a novel and efficient method for the in situ preparation of organic/inorganic composite gels from concentrated brine.

### 2.5. Thermal Stability

Figure 5 presents the thermogravimetric (TG), differential scanning calorimetry (DSC), and differential thermogravimetric (DTG) curves for FG, BG, and CG samples. The thermal analysis is divided into low-, medium-, and high-temperature ranges based on the weight loss observed in the DTG curves, as illustrated in Figure 5. The primary changes within each temperature range are summarized as follows. In the low-temperature range, adsorbed water is desorbed from the gel samples [16,17,62]. Within the medium-temperature range, imidization reactions occur through pendant amide groups [16,62,63,64], marking the onset of significant structural changes in the polymer. In the high-temperature range, processes such as polymer backbone cleavage, gel crosslinking bond breakage, and inorganic decomposition are evident [9,62,63,64]. The thermal decomposition temperature serves as a critical threshold at which the gel samples start to lose thermal stability.

In Figure 5a, the FG-95 sample shows a weight loss of approximately 4% in the temperature range of 30–110 °C, with the weight loss concentrated around 74 °C. This process corresponds to the removal of adsorbed water from the polymer gel, which is consistent with the results reported in reference [65]. For the BG-T15-95 sample (Figure 5b), dehydration spans from 30 to 143 °C, featuring notable weight losses at around 57, 85, and 116 °C. The weight loss at 116 °C corresponds to the removal of adsorbed water from magnesium salts in the brine gel [66], indicating a higher dehydration temperature relative to FG, with a total weight loss of 7.7%. In the case of the CG-T15-U1.00-95 sample (Figure 5c), dehydration occurs between 30 and 134 °C, with the maximum weight loss rate peaking around 95 °C, leading to a weight loss of approximately 2.8%. This lower weight loss suggests a reduced hydrophilicity of the inorganic components in the composite gel. The DSC curves for all gel samples during the dehydration process exhibit endothermic behavior, reflecting the energy absorbed during these transformations.

The imidization reactions for FG, BG, and CG samples occur within moderate temperature ranges of 110–446 °C, 143–738 °C, and 134–761 °C, respectively. The DSC curves for these processes exhibit exothermic behavior. Side-chain decomposition in FG, BG, and CG predominantly takes place at 357 °C, 423 °C and 427 °C. The notably higher thermal decomposition temperatures of BG and CG compared to FG suggest an improvement in thermal stability attributed to the incorporation of inorganic materials. The XRD test results indicate that magnesium ions in the composite gel form basic magnesium carbonate with crystallization water in the presence of urea [52]. In Figure 5c, the composite gel sample exhibits a noticeable weight loss around 214 °C, which corresponds to a typical endothermic peak. This process is associated with the thermal decomposition of magnesium carbonate losing its crystallization water, consistent with the results reported in reference [67].

In the high-temperature range from 446 to 900 °C for FG, 738–900 °C for BG, and 761–900 °C for CG, the DSC curves show endothermic events corresponding to the decomposition of the polymer main chain, rupture of crosslinking bonds within the gel, and decomposition of inorganic components. While FG exhibits substantial decomposition around 637 °C, nearly reaching completion by 900 °C (Figure 5a), BG and CG undergo significant decomposition only beyond their respective onset temperatures of 738 °C and 761 °C (Figure 5b,c). Consequently, due to the presence of inorganic constituents, BG and CG retain residual masses of 59.6% and 57.7%, respectively, at 900 °C.

## 3. Conclusions

This study innovatively employs urea as a precipitating agent to synthesize organic/inorganic composite gels, with low molecular weight PAM as the primary agent, HQ and formaldehyde serving as crosslinking agents. The decomposition of urea not only inhibits nucleophilic substitution reactions, thereby extending the gelation time, but also promotes the in situ transformation of destabilizing calcium and magnesium ions into stable inorganic particles such as calcium carbonate, magnesium carbonate, and basic magnesium carbonate. Consequently, the interplay between the organic and inorganic components results in a robust three-dimensional composite structure that significantly enhances the mechanical strength and stability of the gel.

The gelation time for these composite materials can be precisely controlled within a range of 6.6 to 14.1 days at 95 °C and 2.9 to 6.5 days at 120 °C, with stability durations reaching 155 days and 135 days, respectively. This work pioneers an efficient method for the in situ preparation of organic/inorganic composite gels, exploiting urea’s unique ability to delay high-temperature crosslinking reactions of PAM while simultaneously addressing the challenge posed by high salinity inorganic salts present in formation waters. The resulting heat-resistant and salt-tolerant composite gels represent a significant advancement in material science, offering valuable insights for profile control and water shut-off operations in high-temperature, high-salinity reservoirs.

The method reported in this study for preparing organic/inorganic composite gels demonstrates good adaptability at high temperatures (95–120 °C) and high mineralization levels (5–20 × 10^4^ mg/L). This adaptability can meet the varying production scale needs for enhancing oil recovery in oil fields. The gelation time, gel strength, and stability of the composite gels can be easily controlled. The polymer, cross-linking agents, and urea used in this method are widely sourced and cost-effective, which can help improve the economic efficiency of oil fields and make it suitable for industrial application.

## 4. Materials and Methods

### 4.1. Materials

Acrylamide (AM, AR, 99.0%, batch number E2107072) was supplied by Shanghai Aladdin Biochemical Technology Co., Ltd (Shanghai, China). The other reagents employed in the experimental process, including potassium persulfate (K_2_S_2_O_8_, batch number 20230116), urea (batch number 20221104), hydroquinone (HQ, batch number 20220905), formaldehyde (37 wt%, batch number 20230322), sodium chloride, calcium chloride, and hexahydrate magnesium chloride, were all of analytical grade and purchased from Sinopharm Chemical Reagent Co., Ltd (Shanghai, China). Deionized water was used to prepare the saline solution and to synthesize polyacrylamide and gel samples.

### 4.2. Polyacrylamide Synthesis

Polyacrylamide was synthesized in the laboratory, and the preparation process was similar to our previous report [47], with some modifications. In a typical procedure, 30 g of AM monomer was dissolved in 160 mL of deionized water and transferred into a 500 mL three-neck round-bottom flask at 70 °C. Nitrogen gas was purged through the solution for 30 min at a flow rate of 50 mL/min to remove dissolved oxygen. Subsequently, 60 mg of K_2_S_2_O_8_ was dissolved in 10 mL of deionized water and added dropwise to the flask to initiate polymerization. The solution was continuously stirred under nitrogen and heated at 70 °C for 4.5 h. Then, 400 mL of deionized water was gradually added to dilute the polymer solution, resulting in a uniform polymer stock liquor with a concentration of 5 wt%. The viscosity-average molecular weight of the synthesized PAM was measured to be 1.40 × 10^6^ g/mol using an Ubbelohde viscometer (produced by Sichuan Shubo (Group) Co., Ltd., Chongzhou, China). The degree of hydrolysis was determined to be 1.3% through titration.

### 4.3. Gel Sample Preparation

Sodium chloride, calcium chloride, and magnesium chloride were used to prepare a brine stock solution with a total dissolved solids of 30 × 10^4^ mg/L, utilizing deionized water. After dilution, brine solutions with different mineralization levels were produced to simulate formation water. The composition of the brine stock solution is outlined in Table 7.

In preparing the gel reaction solution, the polymer, brine stock solutions, and deionized water were combined in specified proportions to ensure that the sample’s salinity ranged from 5 to 20 × 10^4^ mg/L [12,48], while the polyacrylamide concentration was maintained between 0.5 and 2.0 wt%. Crosslinking agents, including formaldehyde (0.15–0.40 wt%), HQ (0.16–0.42 wt%), and thiourea stabilizer (0.05 wt%), were subsequently incorporated. Urea was then added and mixed thoroughly, resulting in a gel reaction solution, commonly referred to as gelant. After preparing the solution sample, 20 mL of the gelant was accurately transferred into an ampoule using a disposable syringe. Since thiourea was added to the sample as a stabilizer, primarily to remove oxygen in the solution, an inert gas atmosphere was not used during the sample encapsulation process. The ampoule was sealed using an alcohol blast burner, and the sample was then placed in an oven for constant temperature aging. Regular observations and tests were conducted.

### 4.4. Structure Characterization and Morphology Observation

The polymer samples and freshwater gel samples were freeze-dried separately and then crushed. They were subsequently mixed with KBr and ground at a mass ratio of 1:100 (polymer to KBr) before being pressed into pellets. The resulting samples were analyzed using a Bruker Vertex 70 Fourier Transform Infrared Spectrometer (manufactured by Bruker Corporation, Billerica, Massachusetts, United States) over a wavenumber range of 400 to 4000 cm^−1^.

To analyze the structure of the inorganic components in the composite gel, a brine solution with a mineralization degree of 15 × 10^4^ mg/L was used. Urea was added in an amount equivalent to that of the composite gel with *N* = 1, excluding polymers and crosslinking agents. The sample was heated at 95 °C for 30 days to ensure adequate precipitation. Following washing with water, drying, and grinding, X-ray diffraction (XRD) analysis was conducted using a Japan Rigaku X-ray diffractometer (D/MAX-2500/PC) (manufactured by Rigaku Corporation, Tokyo, Japan) equipped with a Cu Kα radiation source (40 kV, 150 mA, λ = 1.54051 Å) to examine the crystalline structure of the inorganic components. The analysis was performed over a diffraction angle range of 2θ = 5-70 °, with a scanning speed of 10 °/min.

The morphology of the samples was examined using a field emission scanning electron microscope (JSM-6700F) from JEOL Ltd (Tokyo, Japan), facilitating a comparative analysis of the morphological characteristics of BG and CG. Due to the significant presence of inorganic salts in both BG and CG samples, inorganic salt particles precipitated from the liquid phase and adhered to the surface of the polymer during the direct observation of the gel samples. Consequently, this study also included washing the BG and CG samples with water to remove soluble inorganic salts. Afterward, the samples were freeze-dried and coated with gold for surface treatment, allowing for morphological observations to compare and analyze the characteristics of the BG and CG samples before and after washing.

### 4.5. Gelation Performance Observation

The evaluation of gel strength, gelation time, and stability time was conducted following the gel strength code method [8,57,58,59]. The procedure involved sealing 20 mL of gelant in an ampoule and placing it in a temperature-controlled oven for heating. The gel strength was observed and the corresponding code was recorded at certain time.

### 4.6. Rheological Testing

Dynamic rheological performance tests of FG, BG, and CG were conducted using a parallel plate rotational rheometer (ARES-G2) from TA Instruments, USA (New Castle, DE, USA). The primary focus was on the measurement of the elastic modulus (G′) and viscous modulus (G″), as well as the influence of factors such as urea concentration and salinity on the strength of the composite gel. During the experiment, the oscillation frequency was initially set at 1 Hz (6.28 rad/s) for strain scanning to identify the linear viscoelastic region of the sample, where the complex modulus remains constant with shear stress. Following this, frequency scanning was carried out within the established linear viscoelastic region, with oscillation frequencies ranging from 0.01 to 10 Hz, to determine G′ and G″, while maintaining the experimental temperature at 25 °C.

### 4.7. Thermal Stability Testing

Thermogravimetric and differential scanning calorimetry (TG-DSC) analyses of the FG, BG, and CG samples were performed using a PerkinElmer STA 8000 thermal analyzer (manufactured by PerkinElmer, Inc., Waltham, MA, USA) to assess the thermal stability of the dried gel samples. During the experiment, the samples were heated in a nitrogen atmosphere at a rate of 10 °C/min, with the temperature range set between 30 and 900 °C.

## Data Availability

The original contributions presented in this study are included in the article. Further inquiries can be directed to the corresponding authors.

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
