# Peer review of "Urea Delays High-Temperature Crosslinking of Polyacrylamide for In Situ Preparation of an Organic/Inorganic Composite Gel"

_gels, 2025, doi:10.3390/gels11040256_

Round 1
Reviewer 1 Report
Comments and Suggestions for Authors
This paper is devoted to the development of a method of cross-linked polyacrylamide nanocomposite gel allowing to control gelation time and gel stability. The manuscript contains interesting results and is worthy of publishing after minor revision.
Some shortcomings that need to be corrected prior to publication to improve the article are as follows:
Line 108 "amine nitrogen atom" – evidently, "amide nitrogen atom"
Line 166. "Gel strength was recorded according to the Sydansk code method" – As for me, this phrase is superfluous in this place. It is repeated at the beginning of Section 2.3 and Section 4.5 in "Materials and methods"
Fig. 2 legend. " c, e (magnified images after washing)" – evidently, "c, f (magnified images after washing).
High-valent metal ions. There are used only two-valent metal ions.
Table 2. a) The authors wrote that the observation was stopped when the dehydration reached 5 mL. But what volume of the samples was used? I recommend to show the amount of water released as a percentage of the sample weight. If the sample volume was 20ml as written later in the text, indicate this volume in the table legend or in Materials and methods.
b) what concentrations of HQ and HCHO was used to prepare the samples in Table 2 and 3?
Line 293 -301. I am not sure that it is necessary to duplicate so literally the data of Table 4 in the text.
Line 324-325. I do not understand exactly what does it mean "…the gel structure formed at low polymer concentration (0.5 wt%) and low crosslinking agent concentration (0.06 wt%, 0.07 wt%) is incomplete and susceptible to dehydration". Table 5 shows that at low cross-linker concentration (Sample 1 in the Table) ts is rather high - 69 days. Why do the authors say that the gel structure is incomplete?
Line 353. "The findings reveal that by adjusting the N value between 0.1 and 2.0, the mass percentage of urea in the gel reaction fluid varies from 0.14 to 2.71 wt%". – What findings? This is simple calculation. Please change or eliminate this sentence.
Line 459 – 460. The authors stated that the slight increase in elastic modulus with increasing N value from 1.0 to 1.25 (at 120C) suggests that Curea in the reaction system is already adequate. But Fig. 4a shows that the G' value of CG-T15-U0.50-95 and CG-T15-U0.75-95 are almost the same. And then raises sharply for CG-T15-1.0-95. So there is no direct dependence of G' on urea content. The authors have only two experimental points for T=120C. Why do they think that the optimum (adequate) concentration of urea is reached? I think this conclusion is not approved by the data obtained.
Line 461-463. "CG-T15-U1.00-120 (120 °C, aged for 7.3 days, CPAM = 1.0 wt%, CHQ = 0.20 wt%, CHCHO = 0.21 wt%) is lower than that of the CG sample CG-T15-U1.00-95 depicted in Figure 4a (95 °C, aged for 44.7 days, CPAM = 1.0 wt%, CHQ = CHCHO = 0.16 wt%)". But the legend to Table 4 shows: "At 95 °C, CPAM = 1.0 wt%, CHQ = 0.20 wt%, CHCHO = 0.21 wt%ï¼›At 120 °C, CPAM = 1.0 wt%, CHQ = CHCHO = 0.16wt %." Besides, in Line 465 we read: "At 120 °C, the concentration of crosslinking agents is lower". The authors should check this paragraph to avoid inconsistencies between different parts of the text.
Line 463-470. I do not understand what did the authors want to say with these sentences, especially taking into account the inconsistencies between the description of reagent concentration in diverse sentences. Can you rephrase or explain in more details this part of the text?
Figure 5. Please add to the figure legend the data of gel composition. Are there the same concentrations of cross-linker and salinity?
Line 503. What dehydration occurred at 30-110C (4% weight loss) if the dried samples were studied?
Line 521. Did the authors mean that the additional weight loss around 214 °C is noted only for CG gel due to dehydration of basic magnesium carbonate because this carbonate is formed only in the presence of urea? Please add the explanation to the text.
Materials and methods
Line 566. "The viscosity-average molecular weight of the synthesized PAM was measured to be 1.40×106" – Please add 1,4*106 g/mol or Da.
Line 571. Why, when describing the preparation of the gel, do the authors indicate that deionized water was used, but there is no such indication for the synthesis of PAM? As for me this
4.3. Gel sample preparation – this section needs to be expanded. How were the samples heated? In sealed ampoules? What atmosphere was used (inert gas or not)?
Regarding English, I would recommend a few corrections:
Line 154. "phenolic formaldehyde crosslinkers" – I understand what the authors mean but, in my opinion, "hydroquinone/formaldehyde crosslinker" sounds better
Line 314. "These result in over-crosslinking…" – "These reactions result in over-crosslinking"
Author Response
Comments 1: Line 108 "amine nitrogen atom" – evidently, "amide nitrogen atom"
Response 1: According to the scientific suggestions from the reviewer, we have revised the term "amine nitrogen atom" to "amide nitrogen atom" in line 114 of the revised text. Additionally, we have made similar modifications to the revised manuscript at Line 77.
Comments 2: Line 166. "Gel strength was recorded according to the Sydansk code method" – As for me, this phrase is superfluous in this place. It is repeated at the beginning of Section 2.3 and Section 4.5 in "Materials and methods"
Response 2: In accordance with the reviewer's suggestion, the sentence "Gel strength was recorded according to the Sydansk code method" in Line 166 of the original manuscript has been deleted in the revised version. Additionally, the related text in Section 4.5 has been refined. Specifically, the sentence "The evaluation of gel strength, gelation time, and stability time was conducted following the Gel Strength Codes (GSC) method as proposed by Sydansk et al.," has been revised to "The evaluation of gel strength, gelation time, and stability time was conducted following the gel strength code method" (Line 654-655 in the revised manuscript) to avoid excessive repetition. Consequently, the reference numbers 47-49 and 50-59 in the original text have been adjusted to 57-59 and 47-56, respectively.
Comments 3: Fig. 2 legend. " c, e (magnified images after washing)" – evidently, "c, f (magnified images after washing).
Response 3: In the revised manuscript (Line 246 in the revised manuscript), the "c, e" in the Fig. 2 legend has been corrected to "c, f". We appreciate the reviewer's meticulous review.
Comments 4: High-valent metal ions. There are used only two-valent metal ions.
Response 4: The expression of "high-valent metal ions" in the original manuscript has been changed to "divalent metal ions" in the revised manuscript (Line 60, 156, 278 in the revised manuscript).
Comments 5: Table 2. a) The authors wrote that the observation was stopped when the dehydration reached 5 mL. But what volume of the samples was used? I recommend to show the amount of water released as a percentage of the sample weight. If the sample volume was 20ml as written later in the text, indicate this volume in the table legend or in Materials and methods.
- b) what concentrations of HQ and HCHO was used to prepare the samples in Table 2 and 3?
Response 5: a) The description regarding the gel dehydration and the cessation of observation presents the same issue in both Table 2 and Table 3. We have made modifications according to the reviewer's comments. In the revised manuscript, we have added a note in the footnotes of Table 2 and Table 3 specifying that the gel volume is 20 mL.
The footnotes of Tables 2 and 3 have been modified as follows:
* The superscript number on the right of the gel strength symbol represents the dehydration volume (in mL) of a 20 mL gel sample. Stop observation when the dehydration volume reaches 5 mL. (Line 290-291, 293-294 in the revised manuscript)
- b) Firstly, in the revised manuscript, we clearly state that the concentrations of PAM, HQ, and HCHO (CPAM, CHQ, and CHCHO) in the gel samples are provided, and we have added the following sentence: “Concentrations of PAM (CPAM), HQ (CHQ), and HCHO (CHCHO) for the gel samples are also given.” (Line 295-296 in the revised manuscript)
Secondly, we have also specified in the title of Table 2 that CPAM, CHQ, and CHCHO are 1.0 wt%, 0.20 wt%, and 0.21 wt%, respectively; and in the title of Table 3, we have indicated that CPAM, CHQ, and CHCHO are 1.0 wt%, 0.16 wt%, and 0.16 wt%, respectively.
The legends of Table 2 and Table 3 have been revised into the following expression:
Table 2. Gel sample strength code at 95°C across different time intervals (CPAM = 1.0 wt%, CHQ = 0.20 wt%, CHCHO = 0.21 wt%). (Line 289 in the revised manuscript)
Table 3. Gel sample strength code at 120°C across different time intervals (CPAM = 1.0 wt%, CHQ = CHCHO = 0.16wt %). (Line 292 in the revised manuscript)
Comments 6: Line 293 -301. I am not sure that it is necessary to duplicate so literally the data of Table 4 in the text.
Response 6: Thank you for the reviewer's suggestion. Based on the specific data presented in Table 4, we have provided a concise description in the text regarding the relevant content to avoid excessive repetition and enhance the readability of the article. The modified content for the original manuscript from Lines 293-301 is as follows:
The experimental results indicate that the FG samples are characterized by a longer stability time, exceeding 176 days and 126 days at 95 °C and 120 °C, respectively. For the BG samples, as the salinity increases from 5×104 mg/L to 20×104 mg/L, both the gelation time and stability time are significantly shortened. For example, at 95 °C, the tD of BG decreases from 12.4 days to 7.9 days, while the tS drops from 42.0 days to 24.5 days. (Line 308-313 in the revised manuscript)
Comments 7: Line 324-325. I do not understand exactly what does it mean "…the gel structure formed at low polymer concentration (0.5 wt%) and low crosslinking agent concentration (0.06 wt%, 0.07 wt%) is incomplete and susceptible to dehydration". Table 5 shows that at low cross-linker concentration (Sample 1 in the Table) ts is rather high - 69 days. Why do the authors say that the gel structure is incomplete?
Response 7: Table 5 presents the tD and tS data of the composite gel under different conditions of CPAM, CHQ, and CHCHO. Under conditions of low polymer concentration (CPAM = 0.5 wt%) and low crosslinker concentrations (CHQ = 0.06 wt%, CHCHO = 0.07 wt%), the stability times of the composite gels are 37.8 days and 69.2 days, respectively. However, the structures of these gel samples remain incomplete and are prone to dehydration, as their maximum gel strengths correspond to the codes of H and F, which do not reach the gel strength code of I. We have provided explanations regarding the strengths of the aforementioned gels in the revised manuscript.
Lines 319-325 in the original manuscript have been revised into the following sentences: (Line 332-340 in the revised manuscript)
At 95 °C, we investigated the effects of CPAM, CHQ, and CHCHO on tD and tS of the composite gel, which had a salinity of 15×104 mg/L and a fixed Curea of 0.68 wt% (N=0.5). The experimental data are summarized in Table 5, where CPAM, CHQ and CHCHO are maintained ranging from 0.5 to 2.0 wt%, 0.06 to 0.31 wt%, and 0.07 to 0.33 wt%, respectively. Under conditions of low polymer concentration (CPAM=0.5 wt%) and low crosslinker concentrations (CHQ=0.06 wt%, CHCHO=0.07 wt%), the stability times of the composite gels are found to be 37.8 days and 69.2 days, respectively. However, the structures of these gel samples remain incomplete and susceptible to dehydration, as their maximum gel strengths correspond to the codes of H and F, rather than the code of I.
Comments 8: Line 353. "The findings reveal that by adjusting the N value between 0.1 and 2.0, the mass percentage of urea in the gel reaction fluid varies from 0.14 to 2.71 wt%". – What findings? This is simple calculation. Please change or eliminate this sentence.
Response 8: We agree with the reviewer's opinion and have deleted the relevant statements in the original text (Line 353-355 in the original manuscript).
Comments 9: Line 459 – 460. The authors stated that the slight increase in elastic modulus with increasing N value from 1.0 to 1.25 (at 120C) suggests that Curea in the reaction system is already adequate. But Fig. 4a shows that the G' value of CG-T15-U0.50-95 and CG-T15-U0.75-95 are almost the same. And then raises sharply for CG-T15-1.0-95. So there is no direct dependence of G' on urea content. The authors have only two experimental points for T=120C. Why do they think that the optimum (adequate) concentration of urea is reached? I think this conclusion is not approved by the data obtained.
Response 9: We agree with the reviewers' point that the relevant expressions in the original manuscript were not sufficiently rigorous. Therefore, we first revised Figure 4a and Figure 4b in the manuscript. In Fig. 4a, we added the G’ and G’’ versus oscillation frequency curves for the CG-T15-U1.25-95 and CG-T15-U1.50-95 samples, and in Figure 4b, we included the G’ versus oscillation frequency curve for the CG-T15-U1.50-120 sample. Based on these additions, we made modifications in the revised manuscript regarding the discussion on the effect of urea concentration on the elastic modulus of the composite gel at 95 °C and 120 °C in Figure 4a and Figure 4b.
The revised Figure 4a shows that when the N value is further increased to 1.25-1.50, the G’ values for the composite gels CG-T15-U1.25-95 and CG-T15-U1.50-95 are 103.5-144.0 Pa and 103.8-150.3 Pa, respectively. Compared to the G’ range (94.9-131.2 Pa) of the CG-T15-U1.00-95 sample (N=1.00), they still maintain an upward trend, but the rate of increase gradually slows down. It is analyzed that with the increase of urea concentration (N=0.3-1.0), the reaction between calcium and magnesium ions in the composite gel and the hydrolysis products of urea can generate more inorganic particles, which helps to increase the strength of the composite gel. When the urea content is further increased (N=1.25-1.50), the alkaline environment generated by urea decomposition is unfavorable for the crosslinking reaction; as shown in Figure 3a, high urea concentration conditions can significantly prolong the gelation time of the composite gel. Therefore, at N=1.25-1.50, the increase in the elastic modulus of the composite gel diminishes.
The revised manuscript has modified the content as follows: (Line 453-475 in the revised manuscript)
In Figure 4a, there are six CG samples including CG-T15-U0.30-95, CG-T15-U0.50-95, CG-T15-U0.75-95, CG-T15-U1.00-95, CG-T15-U1.25-95, and CG-T15-U1.50-95. The salinity and polymer concentration were fixed at 15×104 mg/L and 1 wt%, respectively. As the N value increases from 0.30 to 1.50, the elastic modulus ranges for the six CG samples are 64.5-75.6 Pa, 60.0-82.4 Pa, 75.2-90.9 Pa, 94.9-131.2 Pa, 103.5-144.0 Pa, and 103.8-150.3 Pa, respectively. These results indicate that the elastic modulus of CG-T15-U0.30-95 is slightly higher than that of BG-T15-95 (51.3-73.7 Pa). As the N value further increases to 0.50 and 0.75, the strengths of these two composite gel samples rise, showing an increase of 10-15 Pa in elastic modulus compared to that of CG-T15-U0.30-95. When the N value reaches 1.00 in sample CG-T15-U1.00-95, there is a significant increase in gel strength, with an increase of about 20-40 Pa. When the N value is further increased to 1.25-1.50, the elastic modulus G’ of the composite gel continues to show an upward trend, but the increase is relatively small. Under low urea concentration conditions (N=0.3-1.0), as the urea concentration increases, the reaction between calcium and magnesium ions in the composite gel and the hydrolysis products of urea can produce a larger number of inorganic particles, which helps to increase the strength of the composite gel[17,21,26]. Under high urea concentration conditions (N=1.25-1.50), further increasing the urea content creates an alkaline environment due to urea decomposition, which is unfavorable for the PAM crosslinking reaction. As shown in Figure 3a, high urea concentration conditions can significantly prolong the gelation time of the composite gel. Therefore, at N=1.25-1.50, the increase in the elastic modulus of the composite gel diminishes.
The revised Figure 4b shows the elastic modulus G’-oscillation frequency curves of three urea-containing composite gels (CG-T15-U1.00-120, CG-T15-U1.25-120, and CG-T15-U1.50-120) at 120 °C, with N values of 1.00, 1.25, and 1.50, respectively. Under the experimental conditions, the elastic modulus ranges of these three composite gels are 40.2-54.8 Pa, 41.1-56.6 Pa, and 41.7-57.8 Pa, respectively. As the N value increases, the elastic modulus of the composite gels shows a slight increase, with the rate of increase gradually decreasing. This observation is similar to the phenomenon noted in Figure 4a. The influence of urea on the growth of inorganic particles and its effect on the PAM crosslinking reaction jointly determine the elastic modulus of the composite gel. In the revised manuscript, the following sentence has been removed: "This slight increase in elastic modulus suggests that Curea in the reaction system is already adequate."
The content of the original manuscript (Line 455-460) has now been modified to the following: (Line 476-484 in the revised manuscript)
Figure 4b presents the elastic modulus G’-oscillation frequency curves of three urea-containing composite gels (CG-T15-U1.00-120, CG-T15-U1.25-120, and CG-T15-U1.50-120) aged 7.3 days at 120 °C, with N values of 1.00, 1.25, and 1.50, respectively. Under the experimental conditions, the elastic modulus ranges of these three composite gels are 40.2-54.8 Pa, 41.1-56.6 Pa, and 41.7-57.8 Pa, respectively. As the N value increases, the elastic modulus of the composite gels shows a slight increase, with the rate of increase gradually decreasing. This observation is similar to the phenomenon noted in Figure 4a. The influence of urea on the growth of inorganic particles and its effect on the PAM crosslinking reaction jointly determine the elastic modulus of the composite gel.
Comments 10: Line 461-463. "CG-T15-U1.00-120 (120 °C, aged for 7.3 days, CPAM = 1.0 wt%, CHQ = 0.20 wt%, CHCHO = 0.21 wt%) is lower than that of the CG sample CG-T15-U1.00-95 depicted in Figure 4a (95 °C, aged for 44.7 days, CPAM = 1.0 wt%, CHQ = CHCHO = 0.16 wt%)". But the legend to Table 4 shows: "At 95 °C, CPAM = 1.0 wt%, CHQ = 0.20 wt%, CHCHO = 0.21 wt%ï¼›At 120 °C, CPAM = 1.0 wt%, CHQ = CHCHO = 0.16wt %." Besides, in Line 465 we read: "At 120 °C, the concentration of crosslinking agents is lower". The authors should check this paragraph to avoid inconsistencies between different parts of the text.
Response 10: We sincerely appreciate the reviewer's thorough review and examination, and we apologize for our oversight. Based on your suggestion, we have rechecked and corrected the compositions of the CG-T15-U1.00-120 and CG-T15-U1.00-95 samples. The statement in the original manuscript (Line 460-463) has been revised to the following: (Line 484-487 in the revised manuscript)
In Figure 4b, the elastic modulus of the CG sample CG-T15-U1.00-120 (120 °C, aged for 7.3 days, CPAM = 1.0 wt%, CHQ = CHCHO = 0.16 wt%) is lower than that of the CG sample CG-T15-U1.00-95 depicted in Figure 4a (95 °C, aged for 44.7 days, CPAM = 1.0 wt%, CHQ =0.20 wt%, CHCHO = 0.21 wt%).
Comments 11: Line 463-470. I do not understand what did the authors want to say with these sentences, especially taking into account the inconsistencies between the description of reagent concentration in diverse sentences. Can you rephrase or explain in more details this part of the text?
Response 11: As mentioned above, we have corrected the composition of the gel samples. Currently, in the revised manuscript, we have made modifications and provided new explanations in response to the reviewer’s suggestions. The content from lines 463-466 in the original manuscript has been revised as follows: (Line 487-494 in the revised manuscript)
When the composition of the gel agent is the same, high temperature promotes the polymer crosslinking reaction and the formation of inorganic particles, resulting in a composite gel with a faster gelation rate and higher gel strength. To delay the crosslinking reaction of the composite gel under high temperature conditions of 120 °C, we selected a lower concentration of crosslinking agent to prolong the gelation time and control the gel strength. Additionally, since the aging time of the composite gel at 120 °C is relatively short in this study, the elastic modulus of the composite gel samples in Figure 4b is relatively small.
Comments 12: Figure 5. Please add to the figure legend the data of gel composition. Are there the same concentrations of cross-linker and salinity?
Response 12: In accordance with the reviewer's suggestion, we have added the sample composition to the Figure 5 legend. The modified legend is as follows: (Line 543-545 in the revised manuscript)
Figure 5. TG-DSC-DTG curves of FG (a), BG (b) and CG (c). (CPAM = 1.0 wt%, CHQ = 0.20 wt%, CHCHO = 0.21 wt%. For BG and CG preparation, salinity is fixed as 15×104 mg/L,and N is selected as 1.00 to obtain CG.)
Comments 13: Line 503. What dehydration occurred at 30-110C (4% weight loss) if the dried samples were studied?
Response 13: The freshwater gel sample FG-95 contains a large number of amide groups, which have a strong affinity for water. Although the FG-95 samples underwent freeze-drying before thermal analysis testing, the amide groups may still adsorb moisture from the air through hydrogen bonding. In Figure 5a, the FG-95 sample shows a weight loss of approximately 4% in the temperature range of 30-110 °C, corresponding to the removal of adsorbed water from the polymer gel. A new reference [65] has been added, and subsequent reference numbers have been adjusted accordingly. The revised manuscript has made adjustments to the content of Lines 503-505 of the original text. The revised content is as follows: (Line 529-532 in the revised manuscript)
In Figure 5a, the FG-95 sample shows a weight loss of approximately 4% in the temperature range of 30-110 °C, with the weight loss concentrated around 74 °C. This process corresponds to the removal of adsorbed water from the polymer gel, which is consistent with the results reported in reference [65].
Comments 14: Line 521. Did the authors mean that the additional weight loss around 214 °C is noted only for CG gel due to dehydration of basic magnesium carbonate because this carbonate is formed only in the presence of urea? Please add the explanation to the text.
Response 14: We agree with the reviewer's point of view and have modified and supplemented the content of Lines 520-522 of the original manuscript. It states that in the composite gel shown in Figure 5c, magnesium ions form basic magnesium carbonate with crystallization water in the presence of urea [52]. This inorganic compound loses its crystallization water around 214 °C and is associated with a significant endothermic peak, which is consistent with the findings reported in reference [67]. The revised content of the manuscript is as follows: (Line 551-557 in the revised manuscript)
The XRD test results indicate that magnesium ions in the composite gel form basic magnesium carbonate with crystallization water in the presence of urea [52]. In Figure 5c, the composite gel sample exhibits a noticeable weight loss around 214 °C, which corresponds to a typical endothermic peak. This process is associated with the thermal decomposition of magnesium carbonate losing its crystallization water, consistent with the results reported in reference [67].
Materials and methods
Comments 15: Line 566. "The viscosity-average molecular weight of the synthesized PAM was measured to be 1.40×106" – Please add 1,4*106 g/mol or Da.
Response 15: We have revised the manuscript according to the reviewer's suggestions. The revised sentence is as follows: (Line 612-613 in the revised manuscript)
The viscosity-average molecular weight of the synthesized PAM was measured to be 1.40×106 g/mol.
Comments 16: Line 571. Why, when describing the preparation of the gel, do the authors indicate that deionized water was used, but there is no such indication for the synthesis of PAM? As for me this
Response 16: Firstly, in Section 4.1 Materials, we have added a note indicating that deionized water is used for the preparation of the saline solution as well as for the synthesis of polymer and gel samples. Then, in Section 4.2, we specifically clarified that deionized water was used in the preparation process of PAM, in accordance with the revision suggestions. The modified relevant statements are as follows:
4.1. Materials (Line 600-601 in the revised manuscript)
…… Deionized water was used to prepare the saline solution and to synthesize polyacrylamide and gel samples.
4.2. Polyacrylamide synthesis (Line 604-612 in the revised manuscript)
…… In a typical procedure, 30 g of AM monomer was dissolved in 160 mL of deionized water and transferred into a 500 mL three-neck round-bottom flask at 70 °C. Nitrogen gas was purged through the solution for 30 minutes at a flow rate of 50 mL/min to remove dissolved oxygen. Subsequently, 60 mg of K2S2O8 was dissolved in 10 mL of deionized water and added dropwise to the flask to initiate polymerization. The solution was continuously stirred under nitrogen and heated at 70 °C for 4.5 hours. Then, 400 mL of deionized water was gradually added to dilute the polymer solution, resulting in a uniform polymer stock liquor with a concentration of 5 wt%.
Comments 17: 4.3. Gel sample preparation – this section needs to be expanded. How were the samples heated? In sealed ampoules? What atmosphere was used (inert gas or not)?
Response 17: In accordance with the reviewer's suggestions, the revised manuscript has included the following operations in the gel sample preparation section 4.3. (Line 628-633 in the revised manuscript)
After preparing the solution sample, 20 mL of the gelant was accurately transferred into an ampoule using a disposable syringe. Since thiourea was added to the sample as a stabilizer, primarily to remove oxygen in the solution, an inert gas atmosphere was not used during the sample encapsulation process. The ampoule was sealed using an alcohol blast burner, and the sample was then placed in an oven for constant temperature aging. Regular observations and tests were conducted.
Regarding English, I would recommend a few corrections:
Comments 18: Line 154. "phenolic formaldehyde crosslinkers" – I understand what the authors mean but, in my opinion, "hydroquinone/formaldehyde crosslinker" sounds better
Response 18: In accordance with the revision suggestion, the manuscript has been modified to change "phenolic formaldehyde crosslinkers" to "hydroquinone/formaldehyde crosslinker." (Line 163, 269, 326, 362 in the revised manuscript)
Comments 19: Line 314. "These result in over-crosslinking…" – "These reactions result in over-crosslinking"
Response 19: We have revised the relevant content accordingly, (Line 326 in the revised manuscript) which has enhanced the accuracy of the article's presentation.

Reviewer 2 Report
Comments and Suggestions for Authors
- Please include a sentence in the abstract about the urea-based gel's efficiency and performance compared to other methods to showcase the approach's innovative nature while pointing out how it performs better than conventional methods.
2. The introduction presents an extensive examination of current challenges in polymer gel applications for oilfields while examining how high temperatures and salt content affect gelation time and steadiness. The analysis should define how these problems hinder technology scalability and application in different geological settings. Please justify the need for innovative solutions like organic/inorganic composite gels.
3. The introductory section points out the limitations of polymer and crosslinker concentration adjustments while stressing the importance of understanding basic reaction mechanisms. Researchers should refer to existing significant studies and discoveries that attempted to fill this research gap while also discussing their methodologies' limitations. Please demonstrate the effectiveness of the proposed strategy to solve current problems, specifically with composite gel synthesis in high-salinity conditions.4. Please provide a concise evaluation of the method's scalability potential for industrial use.
5. The authors specify in Section 4.1 that they obtained the reagents from Sinopharm Chemical Reagent Co., Ltd. and confirm these reagents meet analytical grade standards. Understanding whether the purity of reagents underwent verification before use would be beneficial since experimental results can be altered by impurities. The experimental procedure's reproducibility and transparency would benefit from a brief note on batch numbers or lot numbers of essential reagents like K2S2O8 and formaldehyde.
6. Section 4.2 presents the polyacrylamide synthesis procedure with specifications for temperature requirements and details about nitrogen flow rates and polymerization duration. What was the reason behind purging nitrogen for 30 minutes at 50 mL/min flow rate? The nitrogen flow rate and duration of polymerization time may impact the polymer properties like molecular weight and degree of hydrolysis. A deeper understanding of future research can be achieved by examining the effects of these parameters on the synthesis process.
7. The authors explain how they create brine stock solutions with different mineralization concentrations to represent formation water in Section 4.3. The authors need to explain how they chose the specific ion concentrations (Na+, Ca2+, Mg2+, Cl-) and total dissolved solids to match real-world formation water conditions despite this information being listed in Table 7. The research targets specific geological formations or reservoirs for investigation. Providing a more detailed explanation would increase the practical applicability of the experimental setup.
8. The researchers measured gelation performance with Gel Strength Codes while conducting rheological testing with a parallel plate rotational rheometer. The authors fail to mention if they performed pre-testing or calibration on the rheometer for composite gels along with information about potential time-dependent variations in gelation and rheological properties after gelation. The paper should describe the duration of curing time for the gelants before testing and examine whether changes in experimental conditions like temperature or humidity impacted the measurements of gelation time or gel strength. Understanding the relationship between the oscillation frequency range (0.01 to 10 Hz) and the viscoelastic properties of composite gels will enhance the interpretation of rheological measurements.
Author Response
Comments 1: Please include a sentence in the abstract about the urea-based gel's efficiency and performance compared to other methods to showcase the approach's innovative nature while pointing out how it performs better than conventional methods.
Response 1: According to the reviewer's suggestion, we have revised the abstract as follows: (Line 13-28 in the revised manuscript)
To address the rapid crosslinking reaction and short stability duration of polyacrylamide gel under high salinity and temperature conditions, this paper proposes utilizing urea to delay the nucleophilic substitution crosslinking reaction among polyacrylamide, hydroquinone, and formaldehyde. Additionally, urea regulates the precipitation of calcium and magnesium ions, enabling the in situ preparation of an organic/inorganic composite gel consisting of crosslinked polyacrylamide and carbonate particles. With calcium and magnesium ion concentrations at 6817 mg/L and total salinity at 15×104 mg/L, the gelation time can be controlled to range from 6.6 to 14.1 days at 95 °C and from 2.9 to 6.5 days at 120 °C. The resulting composite gel can remain stable for up to 155 days at 95 °C and 135 days at 120 °C. The delayed gelation facilitates longer-distance diffusion of the gelling agent into the formation, while the enhancements in gel strength and stability provide a solid foundation for improving the effectiveness of profile control and water shut-off in oilfields. The urea-controlling method is novel and effective in extending the high-temperature cross-linking reaction time of polyacrylamide. By converting calcium and magnesium ions into inorganic particles, it enables the in-situ preparation of organic/inorganic composite gels, enhancing their strength and stability.
Comments 2: The introduction presents an extensive examination of current challenges in polymer gel applications for oilfields while examining how high temperatures and salt content affect gelation time and steadiness. The analysis should define how these problems hinder technology scalability and application in different geological settings. Please justify the need for innovative solutions like organic/inorganic composite gels.
Response 2: We agree with the reviewers' suggestion and have supplemented the following statements in the first paragraph of the introduction: (Line 46-50 in the revised manuscript)
In addition, inorganic particles can also serve as multifunctional crosslinkers to form physical crosslinking points with polymer molecules. Organic/inorganic composite gels exhibit improved hydrophilicity, thermal stability, salt resistance, and higher elastic modulus, and have become an important research direction for enhanced oil recovery [20-35].
Comments 3: The introductory section points out the limitations of polymer and crosslinker concentration adjustments while stressing the importance of understanding basic reaction mechanisms. Researchers should refer to existing significant studies and discoveries that attempted to fill this research gap while also discussing their methodologies' limitations. Please demonstrate the effectiveness of the proposed strategy to solve current problems, specifically with composite gel synthesis in high-salinity conditions.
Response 3: Thank you for the suggestion. In fact, in the last paragraph of Section 1. Introduction, we have elaborated on the effectiveness of employing urea to delay the crosslinking reaction of polyacrylamide under high temperature and high salinity conditions. We also discussed the conversion of divalent calcium and magnesium ions into carbonate inorganic particles, which facilitates the construction of organic/inorganic composite gels with crosslinked polyacrylamide. To facilitate the reviewers’ and readers’ understanding of the characteristics of this reaction system, we have made the following revisions and supplement in the revised text: (Line 144-160 in the revised manuscript)
Based on the nucleophilic substitution crosslinking reaction mechanism between PAM and phenolic formaldehyde compounds [38,40], this research employs urea additives to regulate both the gelation time and strength of the composite gel. Urea decomposes at elevated temperatures, producing ammonia and carbon dioxide, which further generate hydroxide and carbonate anions in the solution. These anions inhibit the protonation of the methylol group, thereby extending the gelation time of the gelant. In high-salinity solutions, divalent calcium and magnesium ions interact with carboxylate ions produced from the hydrolysis of amide groups through electrostatic or coordination bonds. Additionally, these calcium and magnesium ions can also react with carbonate or hydroxide ions generated from the decomposition of urea to form inorganic particles, such as carbonates or basic carbonates. These particles subsequently bind closely with polymer molecules, resulting in the formation of high-strength composite gels. This study utilizes divalent metal ions in formation brine as raw materials to in situ synthesize organic/inorganic composite gels. This approach not only mitigates the destabilizing impact of inorganic salts on polymer gels but also enhances the strength and stability of the composite gel through the incorporation of generated inorganic particles, thereby achieving integrated utilization of subsurface mineral resources.
Comments 4: Please provide a concise evaluation of the method's scalability potential for industrial use.
Response 4: We would like to thank the reviewer for the feedback. We have added the following content to the conclusion section of the revised manuscript: (Line 585-591 in the revised manuscript)
The method reported in this study for preparing organic/inorganic composite gels demonstrates good adaptability at high temperatures (95-120 °C) and high mineralization levels (5-20×104 mg/L). This adaptability can meet the varying production scale needs for enhancing oil recovery in oil fields. The gelation time, gel strength, and stability of the composite gels can be easily controlled. The polymer, cross-linking agents, and urea used in this method are widely sourced and cost-effective, which can help improve the economic efficiency of oil field and make it suitable for industrial application.
Comments 5: The authors specify in Section 4.1 that they obtained the reagents from Sinopharm Chemical Reagent Co., Ltd. and confirm these reagents meet analytical grade standards. Understanding whether the purity of reagents underwent verification before use would be beneficial since experimental results can be altered by impurities. The experimental procedure's reproducibility and transparency would benefit from a brief note on batch numbers or lot numbers of essential reagents like K2S2O8 and formaldehyde.
Response 5: Thank you for the reviewer's suggestion. All reagents used in the experiments are of analytical grade, and we conducted multiple experiments to ensure data reproducibility. As per the reviewer's request, we have reconfirmed the manufacturers and batch numbers of the reagents. Upon verification, we found that AM was supplied by Shanghai Aladdin Biochemical Technology Co., Ltd., and we apologize for the carelessness in the original manuscript! We have corrected the information regarding the manufacturer of AM in the revised manuscript, and we have also provided the batch numbers for AM, K2S2O8, urea, HQ, and formaldehyde. The revised content in Section 4.1, Materials, is as follows: (Line 594-601 in the revised manuscript)
Acrylamide (AM, AR, 99.0%, batch number E2107072) was supplied by Shanghai Aladdin Biochemical Technology Co., Ltd.. The other reagents employed in the experimental process, including potassium persulfate (K2S2O8, batch number 20230116), urea (batch number 20221104), hydroquinone (HQ, batch number 20220905), formaldehyde (37 wt%, batch number 20230322), sodium chloride, magnesium chloride, and hexahydrate magnesium chloride, were all of analytical grade and purchased from Sinopharm Chemical Reagent Co., Ltd.. Deionized water was used to prepare the saline solution and to synthesize polyacrylamide and gel samples.
Comments 6: Section 4.2 presents the polyacrylamide synthesis procedure with specifications for temperature requirements and details about nitrogen flow rates and polymerization duration. What was the reason behind purging nitrogen for 30 minutes at 50 mL/min flow rate? The nitrogen flow rate and duration of polymerization time may impact the polymer properties like molecular weight and degree of hydrolysis. A deeper understanding of future research can be achieved by examining the effects of these parameters on the synthesis process.
Response 6: Thank you for the suggestion. In the future research, we will systematically study the effect of nitrogen flow rate and duration on the molecular weight and hydrolysis degree of the polymer. In this paper, we controlled the nitrogen flow rate at 50 mL/min during the preparation of PAM, and the gas was bubbled for 30 minutes before adding the initiator, with the aim of sufficiently removing the dissolved oxygen from the solution. In the future, we will further optimize the nitrogen flow rate and duration according to the reviewer’s recommendations to improve the polymer performance.
Comments 7: The authors explain how they create brine stock solutions with different mineralization concentrations to represent formation water in Section 4.3. The authors need to explain how they chose the specific ion concentrations (Na+, Ca2+, Mg2+, Cl-) and total dissolved solids to match real-world formation water conditions despite this information being listed in Table 7. The research targets specific geological formations or reservoirs for investigation. Providing a more detailed explanation would increase the practical applicability of the experimental setup.
Response 7: To establish the ion concentrations and total dissolved solids (TDS) for our experiments, we referred to both literature data on formation water [12] and simulated formation water data [48]. In particular, literature [12] provided analytical data on the chemical composition of formation water corresponding to the Tahe Oilfield in western China, which has a mineralization of 19.8 × 104 mg/L. This allowed us to ensure our brine solutions closely reflect real-world conditions. Therefore, in the revised manuscript, we have added references in Section 4.3, and the modified results are as follows: (Line 621-625 in the revised manuscript)
In preparing the gel reaction solution, the polymer, brine stock solutions, and deionized water were combined in specified proportions to ensure that the sample's salinity ranged from 5 to 20×104 mg/L[12,48], while the polyacrylamide concentration was maintained between 0.5-2.0 wt%.
Comments 8: The researchers measured gelation performance with Gel Strength Codes while conducting rheological testing with a parallel plate rotational rheometer. The authors fail to mention if they performed pre-testing or calibration on the rheometer for composite gels along with information about potential time-dependent variations in gelation and rheological properties after gelation. The paper should describe the duration of curing time for the gelants before testing and examine whether changes in experimental conditions like temperature or humidity impacted the measurements of gelation time or gel strength. Understanding the relationship between the oscillation frequency range (0.01 to 10 Hz) and the viscoelastic properties of composite gels will enhance the interpretation of rheological measurements.
Response 8: Thank you for your thoughtful comments and suggestions.
Regarding the rheological performance testing of composite gel samples, our laboratory performs regular calibrations of the instruments with PDMS standard samples every three months. We conduct frequency scans to correct the frequency and modulus values at the intersection of the elastic modulus and the viscous modulus, ensuring that the test results fall within an acceptable error range. Therefore, the experimental data we collect is accurate and reliable.
We acknowledge that time-dependent variations in gelation and rheological properties can occur, and future studies may explore these dynamics more thoroughly.
In the revised manuscript, Figure 4 legend describes the curing time for the gelants before testing. The updated Figure 4 legend is as follows: (Line 430-432 in the revised manuscript)
Figure 4. Elastic modulus and viscous modulus of gel samples aged 44.7 d at 95 °C (a), elastic modulus of gel samples aged 7.3 d at 120 °C (b), effects of salinity on the elastic modulus of CG samples aged 44.7 d at 95 °C (c).
We observed and recorded the gelation time of the gel samples using the gel strength code method. Since the gel samples were encapsulated in ampoules and aged in a temperature-controlled chamber, the gelation time of different samples was influenced by the heating temperature. Detailed data and graphs of these results can be found in Tables 2 and 3, as well as in Figure 3.
The rheological properties of the gel samples were measured using a parallel plate rheometer, with the experimental temperature fixed at 25 °C. Each sample was tested for approximately 10 minutes, and the impact of humidity on the samples' rheological behavior was not taken into account during the testing process.
The discussion of the rheological properties of the gel samples in this paper is based on a comprehensive comparative analysis of the storage modulus and loss modulus across a frequency range of 0.01 to 10 Hz. The testing results reflect the primary rheological characteristics of the samples, making the comparative analysis of the related properties reasonable.

Reviewer 3 Report
Comments and Suggestions for Authors
The authors investigate the “Urea Delays High-Temperature Crosslinking of Polyacrylamide for in Situ Preparation of an Organic/Inorganic Composite Gel”. The manuscript is well organized. The authors of this work elaborate the application of urea to delay gelation and improve the performance of polymer in High temperature conditions
An elaborate description of the approach used during the experiment and detailed analysis of the result is described by the author.
Just a couple of comment and suggestions
- It will be good to see the aplication of the composite gel on lab-scale either
- It was observed that the Ca2+ and Mg2+ ions are precipitated by the urea, what is the impact n the rock material.
- You discussed about the behaviour and performance of nanoparticles reaction. you can check out this articles:
- Experimental Study on the Viscosity Behavior of Silica Nanofluids with Different Ions of Electrolytes
- Interfacial Energy for Solutions of Nanoparticles, Surfactants, and Electrolyte
- Mathematical modelling of surface tension of nanoparticles in electrolyte solutionss
Author Response
Comments 1: It will be good to see the application of the composite gel on lab-scale either
Response 1: This is a good suggestion. However, we currently do not have the experimental conditions to conduct research in this area. We hope to carry out core displacement experiments indoors in the future to further observe the application effects of composite gels.
Comments 2: It was observed that the Ca2+ and Mg2+ ions are precipitated by the urea, what is the impact n the rock material.
Response 2: This is indeed a highly valuable question! In this paper, calcium and magnesium ions interact with carboxylate groups generated through the hydrolysis of polymers. The hydrolysis products of urea also engage with calcium and magnesium ions to form inorganic carbonate particles on the polymer surface, which subsequently combine with cross-linked polyacrylamide to create a composite gel. As the polymer gel solution first penetrates into the larger pores of the formation, the composite gel effectively seals high-permeability layers, thereby reducing the heterogeneity of the formation.
In the case of rock materials within the formation, hydroxyl groups are often found on their surfaces. In the presence of urea, calcium and magnesium ions can deposit onto these rock surfaces, forming inorganic particles. Both the particles present on the rock surfaces and those within the composite gel can interact with the cross-linked polymers via electrostatic forces or coordination bonds. Consequently, in the presence of urea, inorganic particles may develop on the surfaces of rock materials, strengthening the interactions with cross-linked polyacrylamide and ultimately improving the strength and sealing properties of the composite gel.
Comments 3: You discussed about the behaviour and performance of nanoparticles reaction. you can check out this articles:
- Experimental Study on the Viscosity Behavior of Silica Nanofluids with Different Ions of Electrolytes
- Interfacial Energy for Solutions of Nanoparticles, Surfactants, and Electrolyte
- Mathematical modelling of surface tension of nanoparticles in electrolyte solutionss
Response 3: We carefully reviewed the three articles recommended by the reviewer, which focus on the viscosity properties, interfacial energies and mathematical modelling of surface tension of nanoparticle electrolyte solutions. In contrast, our study investigates the reaction of calcium and magnesium ions in saline solutions with urea, leading to the formation of inorganic carbonates or basic carbonates. These particles interact with cross-linked polyacrylamide to create an organic/inorganic composite gel, which exhibits delayed gelation reaction, high elastic moduli, and long-term stability, showcasing typical temperature and salt resistance properties. Our research does not address the interactions between nanoparticles or the viscosity and interfacial energy characteristics of nanofluids. Therefore, we did not cite the recommended references in the revised manuscript.

Round 2
Reviewer 2 Report
Comments and Suggestions for Authors
The current version of the manuscript can be accepted for publication
Author Response
Comment: The current version of the manuscript can be accepted for publication.
Response: We would like to express our sincere gratitude for the reviewer’s recognition of our research work. We also appreciate the important revision suggestions made during the review process. The reviewer’s comments have helped us improve the accuracy and scientific rigor of the paper, making it easier to read. Thank you for taking the time to assist us in enhancing our work!